# Exploring Local Memorization in Diffusion Models via Bright Ending Attention

**Chen Chen**[1]    **Daochang Liu**[2]    **Mubarak Shah**[3]    **Chang Xu**[1]

[1]School of Computer Science, Faculty of Engineering, The University of Sydney, Australia
[2]School of Physics, Mathematics and Computing, The University of Western Australia, Australia
[3]Center for Research in Computer Vision, University of Central Florida, USA
`{cche0711@uni., c.xu@}sydney.edu.au daochang.liu@uwa.edu.au shah@crcv.ucf.edu`

## Abstract

Text-to-image diffusion models have achieved unprecedented proficiency in generating realistic images. However, their inherent tendency to memorize and replicate training data during inference raises significant concerns, including potential copyright infringement. In response, various methods have been proposed to evaluate, detect, and mitigate memorization. Our analysis reveals that existing approaches significantly underperform in handling local memorization, where only specific image regions are memorized, compared to global memorization, where the entire image is replicated. Also, they cannot locate the local memorization regions, making it hard to investigate locally. To address these, we identify a novel "bright ending" (BE) anomaly in diffusion models prone to memorizing training images. BE refers to a distinct cross-attention pattern observed in text-to-image diffusion models, where memorized image patches exhibit significantly greater attention to the final text token during the last inference step than non-memorized patches. This pattern highlights regions where the generated image replicates training data and enables efficient localization of memorized regions. Equipped with this, we propose a simple yet effective method to integrate BE into existing frameworks, significantly improving their performance by narrowing the performance gap caused by local memorization. Our results not only validate the successful execution of the new localization task but also establish new state-of-the-art performance across all existing tasks, underscoring the significance of the BE phenomenon.

## 1 Introduction

Text-to-image diffusion models like Stable Diffusion (Rombach et al., 2022) have achieved unparalleled proficiency in creating images that not only showcase exceptional fidelity and diversity but also closely correspond with the user's input textual prompts. This advancement has garnered attention from a broad spectrum of users, leading to the extensive dissemination and commercial utilization of models trained on comprehensive web-scale datasets, such as LAION (Schuhmann et al., 2022), alongside their produced images. However, this widespread usage introduces legal complexities for both the proprietors and users of these models, particularly when the training datasets encompass copyrighted content. The inherent ability of these models to memorize and replicate training data during inference raises significant concerns, potentially infringing on copyright laws without notifying either the model's owners or users or the copyright holders of the replicated content. The challenge is further exacerbated by the training datasets' vast size, which makes thorough human scrutiny unfeasible. Illustratively, several high-profile lawsuits (Saveri & Matthew, 2023) have been initiated against entities like Stability AI, DeviantArt, Midjourney, and Runway AI by distinguished artists. These lawsuits argue that Stable Diffusion acts as a '21st-century collage tool', remixing copyrighted works of countless artists used in its training data.

In response to these legal challenges, Carlini et al. (2023) and Somepalli et al. (2023a) proposed similarity metrics to evaluate memorization, and recent efforts (Somepalli et al., 2023b; Wen et al., 2024; Chen et al., 2024) have focused on developing strategies to detect and mitigate memorization, achieving notable success. However, these metrics and strategies adopt a global perspective, comparing entire generated images to training images. We identify a significant gap in these approaches when dealing with cases where only parts of the training image are memorized, which we refer to as local memorization, as opposed to cases where the entire training image is memorized, termed global memorization. Specifically, current methods can be easily tricked in local memorization scenarios by introducing diversity in non-memorized regions, resulting in many false negatives during evaluation

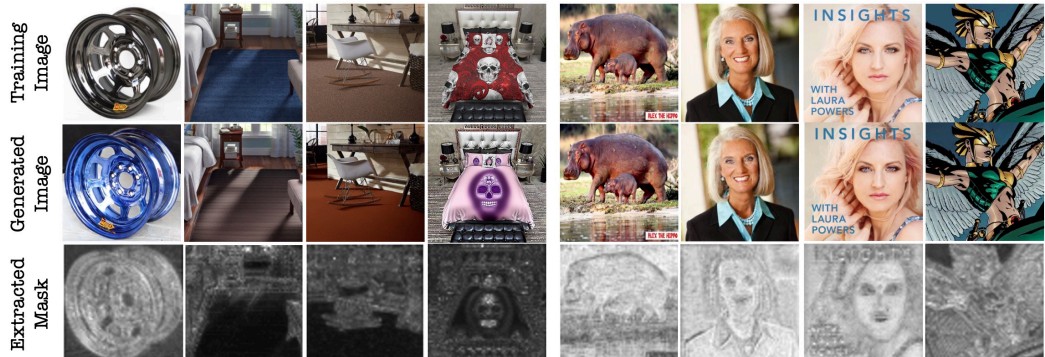

Figure 1: Memorization in diffusion models can occur both globally (right) and locally (left). The first row displays the memorized training images, the second row shows the corresponding generated images, and the third row shows the local memorization masks extracted using 'bright ending', where brightness values quantify the memorization effect. Most pixel values in masks on the right are brighter due to global memorization compared to masks on the left due to local memorization.

and detection. Consequently, this failure to correctly identify local memorization may prevent the activation of mitigation strategies. Motivated by this research gap, we propose a novel view of the memorization issues, where we argue that improved metrics and strategies should concentrate exclusively on the locally memorized regions, as these areas pose risks. In contrast, unmemorized parts of the image, which do not present risks, should be completely ignored.

Building on this localization insight, we introduce a new task: extracting localized memorized regions. This task is crucial for better understanding local memorization and developing targeted strategies to reduce the associated performance gap. To accomplish this task effectively and efficiently, we introduce the concept of the 'bright ending' (BE), a phenomenon observed in the cross-attention maps of text-to-image diffusion models. The bright ending occurs when the end text token in the final inference step exhibits abnormally high attention scores on specific image patches in memorized generations compared to non-memorized ones. This distinctive pattern effectively highlights the regions where the model has memorized the training data, requiring only a single inference pass without relying on access to the training data. To our knowledge, this is the first method to achieve such precise and efficient localization, emphasizing the significance of BE. Finally, we propose a simple yet effective method to integrate BE and its extracted local memorization masks into existing state-of-the-art memorization evaluation, detection, and mitigation strategies. Extensive experiments demonstrate that this integration significantly enhances the performance of these existing tasks by narrowing the gap caused by local memorization, further underscoring the contribution of BE.

In summary, our contributions are three-fold: 1) We analyze the performance gap between local and global memorization in existing tasks and introduce a novel localized view of memorization in diffusion models. 2) We propose a new task of detecting localized memorization regions and propose the novel 'bright ending' (BE) phenomenon, which is the first method to successfully solve this task using only a single inference process without requiring access to training data and conducting a nearest neighbor search over all ground truth images. 3) By integrating the local memorization mask extracted using BE into existing evaluation, detection, and mitigation strategies, we refine these approaches to focus on local perspectives, in contrast to the global focus of all existing strategies. This refinement narrows the gap caused by local memorization and sets a new state-of-the-art in these existing tasks. This simple and effective incorporation further demonstrates the power of BE.

## 2 PRELIMINARIES

### 2.1 DIFFUSION MODELS

Diffusion models (Rombach et al., 2022; Ho et al., 2020; Nichol & Dhariwal, 2021) represent a state-of-the-art class of generative models, emerging as the predominant choice for various generation tasks. They entail a two-stage mechanism, beginning with a forward process that incrementally introduces Gaussian noise to an image originating from a real-data distribution $x_0 \sim q(x)$ across $T$ timesteps, culminating in $x_T \sim \mathcal{N}(0, \mathbf{I})$:

$$q(x_t|x_{t-1}) = \mathcal{N}(x_t; \sqrt{1 - \beta_t}x_{t-1}, \beta_t\mathbf{I}), \tag{1}$$

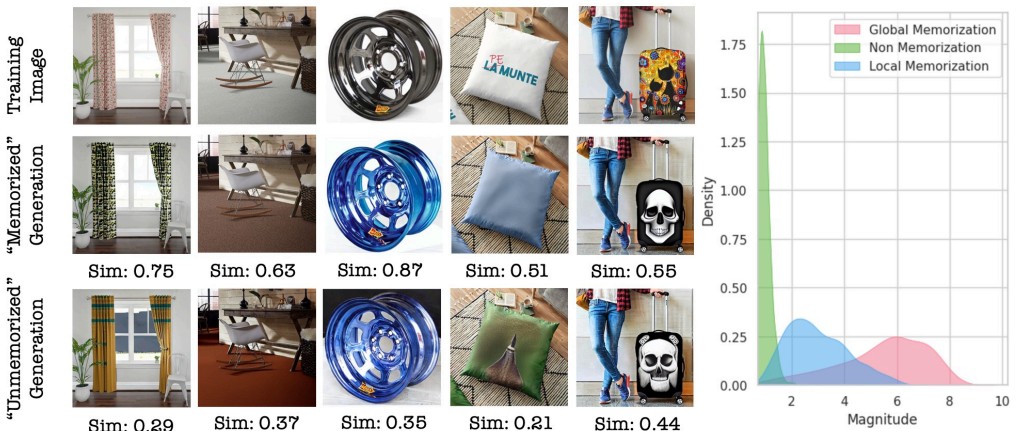

Figure 2: For local memorization cases, variations in non-memorized regions significantly impact existing global similarity and detection measures. Left: Using the current similarity metric SSCD, the second row shows memorized images with similarity scores above 0.5, and the third row shows non-memorized images with scores below 0.5. However, the difference between the second and third rows is minimal, differing only in a local portion, and are, in fact, both local memorization cases, which demonstrates the failure of the global SSCD metric. Right: Currently, magnitude is used to detect memorization, where a higher magnitude corresponds to a greater memorization risk. The distribution reveals a noticeable performance gap: local memorization cases are more challenging to detect than global ones, as their magnitude values overlap more with those of non-memorized cases.

$$q(x_{1:T}|x_0) = \prod_{t=1}^{T} q(x_t|x_{t-1}), \tag{2}$$

where $x_t$ represents the version of $x_0$ with added noise at timestep $t$, and $\{\beta_t\}_{t=1}^{T}$ is the noise schedule controls the amount of noise injected into the data at each step. By using the reparameterization trick (Kingma & Welling, 2014), we can derive the Diffusion Kernel, allowing for the sampling of $x_t$ at any given timestep $t$ in a closed form:

$$q(x_t|x_0) = \mathcal{N}(x_t; \sqrt{\bar{\alpha}_t}x_0, (1 - \bar{\alpha}_t)\mathbf{I}), \tag{3}$$

where $\alpha_t = 1 - \beta_t$ and $\bar{\alpha}_t = \prod_{s=1}^{t} \alpha_s$. Subsequently, in the reverse process, generative modeling is accomplished by training a denoiser network $p_\theta$ to closely estimate the true distribution of $q(x_{t-1}|x_t, x_0)$:

$$p_\theta(x_{t-1}|x_t) = \mathcal{N}(x_{t-1}; \mu_\theta(x_t), \sigma_\theta^2(x_t)\mathbf{I}), \tag{4}$$

$$p_\theta(x_{T:0}) = p(x_T) \prod_{t=T}^{1} p_\theta(x_{t-1}|x_t), \tag{5}$$

where $\mu_\theta(x_t)$ and $\sigma_\theta^2(x_t)$ are the approximated mean and variance of $q(x_{t-1}|x_t, x_0)$, whose true values can be derived using Bayes rule and the Diffusion Kernel as in Eq. 3. The learning objective simplifies to having the denoiser network $\epsilon_\theta$ predict the noise $\epsilon_t$ instead of the image $x_{t-1}$ at any arbitrary step $t$:

$$\mathcal{L} = \mathbb{E}_{t \in [1,T], \epsilon \sim \mathcal{N}(0,\mathbf{I})}[\|\epsilon_t - \epsilon_\theta(x_t, t)\|_2^2], \tag{6}$$

where the denoiser network can be easily reformulated as a conditional generative model $\epsilon_\theta(x_t, y, t)$ by incorporating additional class or text conditioning $y$.

## 3 A LOCALIZED VIEW OF MEMORIZATION IN DIFFUSION MODELS

Fig. 1 illustrates that diffusion models can exhibit both global and local memorization. Global memorization refers to the model remembering the entire image, whereas local memorization pertains to the model retaining only a portion of the image. Notably, global memorization can be seen as a special case of local memorization when the memorized region encompasses the whole image.

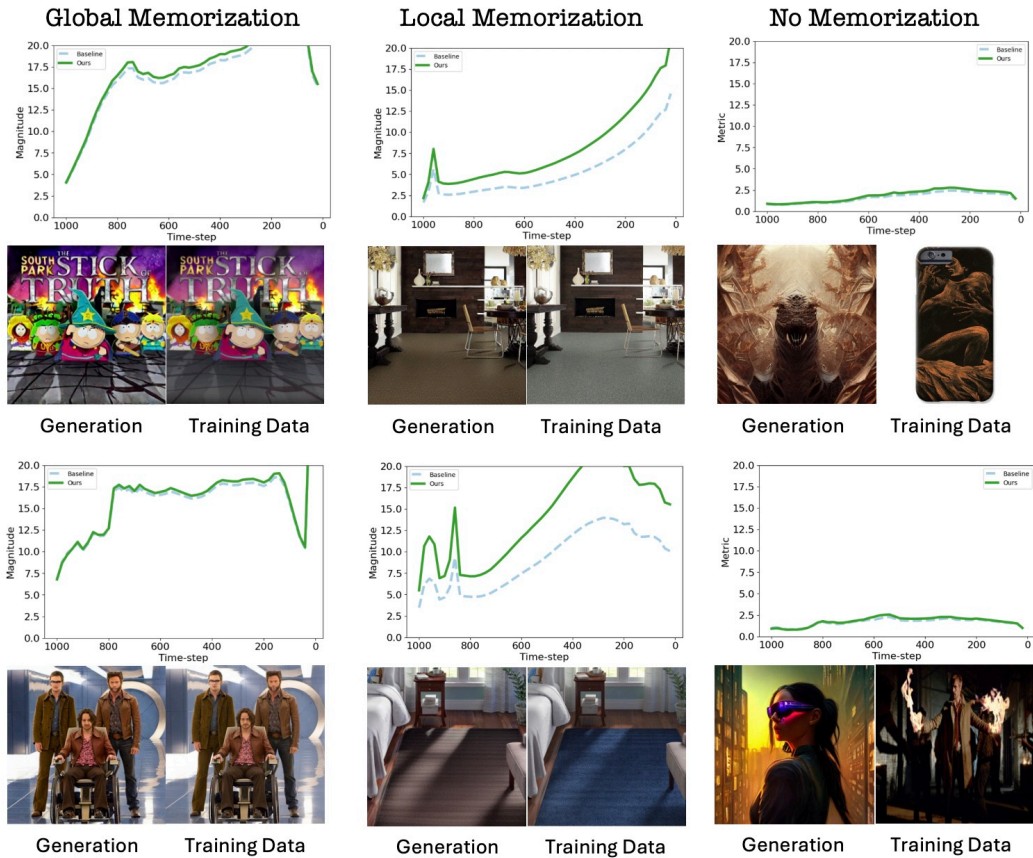

Figure 3: Local memorization tends to have smaller magnitudes than global memorization, making it harder to distinguish from non-memorization. Incorporating BE into the existing detection method effectively increases the magnitude of local cases while keeping the other cases mostly unchanged.

**Existing global metrics fall short for local memorization evaluation.** We observe that all existing studies on memorization in diffusion models are from a global perspective, where the effectiveness of any proposed detection methods and mitigation strategies are evaluated using global metrics, such as the L2 distance or SSCD scores (i.e., cosine similarity of embeddings extracted using the Self-Supervised Copy Detection (SSCD) method (Pizzi et al., 2022)). These metrics measure similarity between entire generated and training images rather than focusing on locally memorized areas. However, these metrics are not robust to variations outside the locally memorized areas and can be easily tricked. Specifically, variations in non-memorized regions can significantly impact these similarity scores despite the similar local memorization regions. A generated image may memorize only a local area of the training image while making the remaining areas novel and diverse, leading to false negatives. Fig. 2 (left) illustrates failures of the global metric approach, where variations in non-memorized regions significantly reduce the similarity scores measured by SSCD. The current standard considers SSCD scores greater than 0.5 as memorization. This results in some images being falsely identified as non-memorized despite all being locally memorized, with differences only in the non-memorized regions.

**Existing global strategies struggle with local memorization detection and mitigation.** Current detection and mitigation strategies are evaluated using global metrics and are thus developed from a global perspective. For instance, Wen et al. (2024) distinguishes memorized cases from unmemorized ones by utilizing the magnitude over the global latent space (Eq. 7):

$$D = \frac{1}{T} \sum_{t=1}^{T} \left\| (\varepsilon_\theta(x_t, e_p) - \varepsilon_\theta(x_t, e_\phi)) \right\|_2, \tag{7}$$

where $e_p$ represents the embedding of the user-provided prompt, $e_\phi$ denotes the embedding of an empty string, $\varepsilon_\theta(x_t, \cdot)$ is the conditional latent-space noise prediction at time $t$, and $D$ is the computed magnitude. The detection method identifies memorized cases by their larger magnitudes

$D$ compared to non-memorized cases. This approach surpasses previous methods, establishing the current state-of-the-art. However, this approach is less practical for local memorization cases than for global ones, as variations in non-memorized regions impact magnitude computations, increasing variance and diminishing the reliability of the detection signal, which leads to more false negatives. To verify this theory, we visualize the density plot of magnitudes for all three scenarios (global, local, and no memorization) in Fig. 2 (right) and Fig. 3 and observed that local cases indeed have smaller magnitudes than global ones, making them closer in magnitude to unmemorized cases and harder to detect. Regarding mitigation, local memorization also underperforms. Wen et al. (2024) employs prompt engineering upon a positive detection, i.e., when $D$ exceeds a threshold $\lambda$, the generation is identified as memorized. Subsequently, $e_p$ is optimized via gradient descent until $D \leq \lambda$, using the global magnitude $D$ as the loss function. Consequently, this optimization process is also sensitive to variations in non-memorized regions. Additionally, the compromised global detection method, which produces more false negatives, often leaves the mitigation strategy untriggered, increasing the risk of unaddressed memorization and potentially leading to legal challenges.

**The localization insight: local memorization is a more generalized and practical notion of memorization.** Recognizing the gap in research on measuring, detecting, and mitigating local memorization, we propose that further investigations should adopt a local perspective. Local memorization is a more meaningful and generalized concept for the following reasons: (1) Unmemorized regions pose no litigation risk and can be disregarded, while even a small locally memorized area can present significant legal concerns. (2) Local memorization is a more encompassing definition, with global memorization being a specific instance. Focusing on local memorization does not diminish the value of global investigations; instead, it complements and extends the scope of existing work.

## 4 BRIGHT ENDING

**Introducing a new task: extracting localized memorized regions.** The observed performance gap caused by local memorization has prompted us to explore a new task: extracting localized memorized regions as a mask. This task is essential for investigating memorization from a local perspective, and the resulting mask can be integrated into existing evaluation, detection, and mitigation strategies. This integration refines these strategies into localized approaches, effectively reducing the performance gap associated with local memorization. One naive method to create such a local memorization mask is to compare generated images with the closest training image. However, this reliance on training data has following limitations: (1) It compromises privacy for a task aimed at preventing the memorization of training data and assumes that the user of the pre-trained model has access to the training data; (2) It requires significantly more computation due to the need for nearest neighbor searches on the large LAION training dataset. Consequently, this naive approach diminishes its practical significance when the mask is used in detection and mitigation algorithms. Therefore, we aim to extract such masks without using training data, relying solely on the pre-trained model's memory to identify distinguishing memorization patterns.

Recent work has highlighted the important role of prompts in causing memorization. For example, Somepalli et al. (2023b) identified that duplicated prompts during the training of text-to-image Stable Diffusion models make the prompts become the "keys" to the models' memory. In other words, when a model encounters a frequently seen prompt, it retrieves and generates images that closely resemble memorized training examples, much like a key unlocking stored content. Similarly, Wen et al. (2024) observed that prompts prone to memorization tend to result in larger text-conditional noise predictions during inference than non-memorization cases. Additionally, Chen et al. (2024) successfully utilized classifier-free guidance to steer generations away from memorized prompts, improving mitigation. Inspired by these findings, we focus on using prompts to extract our desired local mask. This naturally led us to explore the cross-attention mechanism in the U-Net architecture within diffusion models, which links prompts to specific attention-focusing areas in the generated images. One of our key intuitions is that the pre-trained diffusion model's memories resulting from overfitting should differ from those learned through generalization. Memorized generations are inflexible and follow a fixed denoising trajectory that is observably different from non-memorized scenarios. This is evidenced by Wen et al. (2024), who observed that in memorized cases, the magnitude of text-conditional noise prediction during the denoising trajectory is abnormally higher than in non-memorization cases.

**Automatic local memorization mask extraction via bright ending (BE).** Inspired by our intuition, we visualize each token's cross-attention maps in memorized and non-memorized prompts to identify distinguishing patterns (Fig. 4). We discover an interesting phenomenon in pre-trained text-to-image Stable Diffusion models: during the final denoising step, the end token typically shows a dark cross-attention map. Specifically, these dark maps often contain only extremely low-attention finer

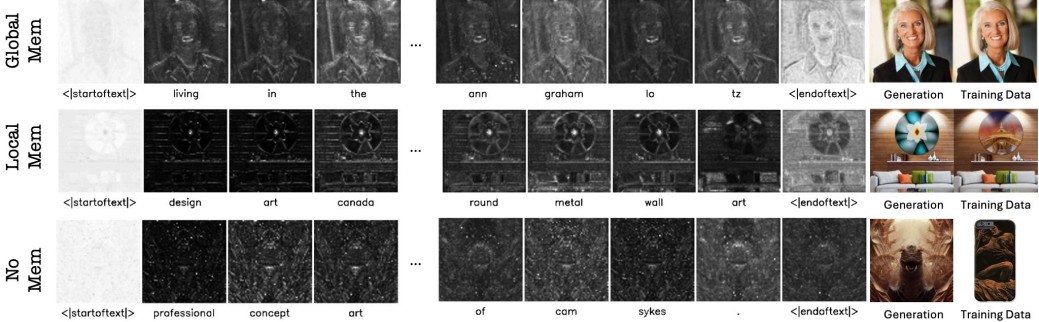

Figure 4: Visualization of cross-attention maps during the final denoising step in pre-trained Stable Diffusion models. Typically, the end token shows a dark cross-attention map, shifting the denoiser's attention from semantic meanings to fine details. However, in memorized models, '**bright ending**' anomaly occurs, where the end token displays abnormally high cross-attention scores, focusing on coarser structures, specifically, the local memorized regions, effectively serving as an efficient automatic extraction of the local memorization mask without needing access to the training data.

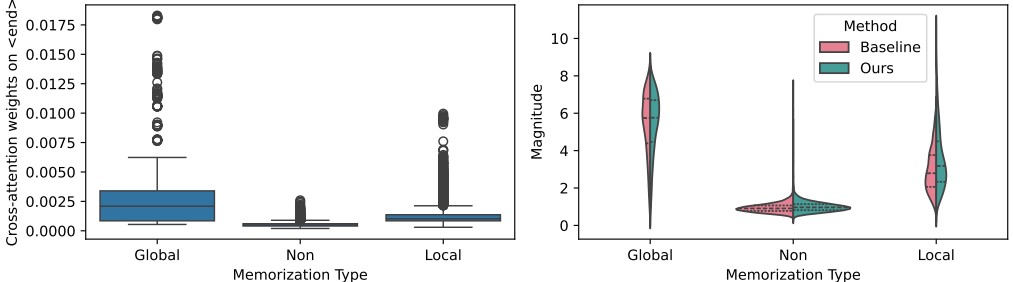

Figure 5: Left: Box plot showing the distribution of end token cross-attention scores during the final inference step for different memorization types. The plot is based on 9,600 generated images, with 16 images generated for 300 memorized and 300 non-memorized prompts each. It clearly distinguishes the three types: non-memorized cases have attention scores close to zero, while memorized cases exhibit abnormally high scores. Global memorization shows higher scores than local memorization due to the larger memorized regions. This further validates the 'bright ending' observation. Right: Violin plots displaying the impact on magnitudes (detection signal) for different memorization types.

edges or random scatter points. Since the end token summarizes all semantic meanings in the prompt, this indicates that when the noised image is very close to a clean image, the denoiser's attention on text-conditioning shifts from focusing on its semantic meanings to concentrating only on fine details.

However, when the model exhibits memorization, we observe an anomaly where the end token displays abnormally high cross-attention scores at the end of denoising steps — a phenomenon we refer to as the 'bright ending' (BE). Specifically, it focuses on the coarser structure of image patches rather than finer details or random scatter points. This indicates that the denoiser remains focused on text-conditioning even during the final inference step, demonstrating overfitting to the text-conditioning. This aligns with the finding in Somepalli et al. (2023b) that memorized generations result from overfitting to repeated captions used as conditioning signals for text-to-image generation during training, thereby following a fixed noise trajectory during inference. Thus, BE is a robust indicator for differentiating memorized generations from unmemorized ones, with bright regions identifying the memorized areas. This effectively functions the BE attention map as an automatic memorization mask, acting as a global mask during global memorization and a local mask during local memorization, as can be visually observed in Fig. 4 and 1. To further validate this observation, we used 300 memorized prompts from Webster (2023) and 300 non-memorized prompts to generate 16 images for each prompt, resulting in a total of 9600 generations. We then employ a box plot (Fig. 5) to visualize the distribution of the attention scores of the end token during the final inference step for these generated images. We observe a clear distinction among the three memorization types: non-memorized cases consistently had attention scores close to zero, while memorized cases exhibited abnormally high scores. Furthermore, global memorization showed higher scores than local memorization due to the larger regions being memorized in global cases compared to local ones.

Beyond BE's effectiveness in performing the new task of extracting localized memorized regions, it is also remarkably efficient, requiring only a single inference pass without needing access to the training data. To our knowledge, BE is the first method to achieve such precise and efficient localization.

## 5 LOCALIZED DETECTION, MITIGATING AND EVALUATION METHODS

### 5.1 INCORPORATING THE BRIGHT ENDING (BE)

BE's success in extracting localized memorized regions can also benefit existing tasks of evaluating, detecting, and mitigating memorization in diffusion models. By integrating the extracted local mask into current strategies, these approaches can adopt a local perspective, thereby reducing the performance gap associated with local memorization. Specifically, as discussed in Sec. 4, the bright ending (BE) attention map inherently serves as an automatically extracted memorization mask. This mask directly utilizes the BE attention scores as its values, eliminating the need for thresholding to convert it into a binary mask of only 0s and 1s. By doing so, it avoids the added variability introduced by thresholding hyperparameters, which may be less effective under certain threshold settings, thereby improving robustness. The approach remains effective regardless of whether the memorized regions are subtle or abstract, as it solely relies on the pre-trained model's overfitted memory during memorization to extract the mask. This eliminates the dependence on factors such as the degree of subtlety or abstraction of memorized regions or thresholding based on size or intensity of memorization. For a qualitative evaluation of the mask, please refer to Fig. 1 and Fig. 4, where brighter patches indicate higher attention scores and darker patches indicate lower ones.

**Detection strategy.**   As discussed in Sec. 3, computing magnitudes in the global space, as shown in Eq. 7, can be misleading during local memorization. To ensure that the magnitude computation corresponds solely to the locally memorized area and remains invariant to other areas, we propose element-wise multiplication of the magnitude by our memorization mask extracted via BE:

$$LD = \frac{1}{T} \sum_{t=1}^{T} \left\| (\varepsilon_\theta(x_t, e_p) - \varepsilon_\theta(x_t, e_\phi)) \circ \mathbf{m} \right\|_2 \bigg/ \left( \frac{1}{N} \sum_{i=1}^{N} m_i \right) \tag{8}$$

Here, $N$ is the number of elements in the mask $\mathbf{m}$. Unlike methods requiring hyperparameter tuning, such as setting thresholds to convert attention scores into a binary mask, our approach uses the attention scores directly as weights. This reweights the relative importance of each pixel in the magnitude computations. To ensure comparability, the result is normalized by the mean of the attention weights $\mathbf{m}$.

**Mitigation strategy.**   Also, as discussed in Sec. 3, the baseline mitigation strategy employs prompt engineering when a positive detection outcome is identified, using global magnitude as the loss function for such prompt optimization. Our proposed BE introduces two key enhancements to this process: (1) *Accurate trigger*: The mitigation strategy is triggered more accurately due to our improved detection method, which utilizes masked local magnitude by incorporating BE. (2) *Improved loss function*: By using masked local magnitude as the loss function, we provide more effective gradient information during backpropagation for prompt optimization, further enhancing the efficacy of the mitigation strategy, again thanks to the incorporation of BE.

**Evaluation strategy.**   To verify the effectiveness of the bright ending when used for evaluating metrics, we design a localized similarity metric, denoted as LS, to measure the similarity between generated image $\hat{x}$ and training image $x$. This metric is a masked version of the L2 distance and is combined with the standard SSCD metric, as defined in Eq. 9. We expect this approach to outperform the sole use of SSCD and the combination of SSCD with the original global L2 distance (denoted as S), as in Eq. 10.

$$LS(\hat{x}, x) = -\mathbb{1}_{\text{SSCD}<0.5} \cdot \left\| (\hat{x} - x) \circ \mathbf{m} \right\|_2, \tag{9}$$

$$S(\hat{x}, x) = -\mathbb{1}_{\text{SSCD}<0.5} \cdot \left\| (\hat{x} - x) \right\|_2 \tag{10}$$

### 5.2 EXPERIMENTS

**Setup.**   Following previous works, we conducted experiments on Stable Diffusion v1-4. We adhered to the baseline prompt dataset (Wen et al., 2024), using 500 prompts each from Lexica, LAION, COCO, and random captions as non-memorized prompts. We used the dataset organized by Webster (2023) for memorized prompts. However, since not all 500 prompts in Webster (2023)'s dataset are prone to memorization, we selected 300 memorized prompts for our experiments. For each memorized and non-memorized prompt, we generated 16 images.

We follow the baseline methodology from Wen et al. (2024) and evaluate performance using the area under the curve (AUC) of the receiver operating characteristic (ROC) curve, the True Positive Rate at 1% False Positive Rate (T@1%F), and the F1 score, and report the detection performance during the 1, 10, and 50 inference steps. However, the baseline detection performance measures whether a prompt is prone to memorization rather than if a specific generation is memorized. They experimented with how well the average magnitude of 1, 4, and 32 generations per prompt differentiates between memorized and non-memorized cases. This approach is flawed because using different random seeds, a prompt prone to memorization can sometimes generate non-memorized images. Consequently, a prompt that rarely yields memorization might still be classified as memorized. For example, generating four non-memorized images from a prompt classified as memorized due to its occasional memorized output can falsely inflate performance metrics. Conversely, failing to classify these as non-memorized when they are all non-memorized should not be penalized. To address this, we assess detection performance based on accurately predicting each generation as memorized or not rather than each prompt. We re-implement the baseline method under the same conditions for a fair comparison. For mitigation, the output utility is evaluated based on the CLIP score, which measures the text-alignment of generated outputs. We evaluate similarity scores using both traditional SSCD and our proposed localized metric to provide a comprehensive performance view, experimenting with multiple privacy levels. Due to current metrics' failure to accurately identify memorized cases, we manually label the ground truth for each generated image from the memorized prompts. Webster (2023) organizes the nearest training images for these generations, making it easy to observe local memorization cases with minimal subjectivity in manual labeling. Please check out Sec. 9.3 for a detailed illustration of the labeling process.

We separately evaluate performance for local and global memorization, contrasting our localized strategies with the baseline's globalized strategies. Table 1 presents a comparison between our localized detection strategy, which uses LD as the detection signal (Eq. 8), and the baseline globalized detection strategy that relies on D (Eq. 7). Figures 6 and 7 highlight comparisons of localized mitigation, utilizing LD (Eq. 8) as both the triggering signal and loss function, with globalized mitigation strategies that use D (Eq. 7). Finally, Table 2 compares the localized similarity metric LS (Eq. 9) against S (Eq. 10) and SSCD in terms of F1-score, using memorization thresholds. For SSCD, we follow the standard threshold of 0.50, while for LS and S, we adopt a threshold of -50.

Implementationally, we experimented with the cross-attention maps on different layers of the U-Net and found that averaging the first two 64-pixel downsampling layers most effectively extracts the local memorization mask. The inference process takes about 2 seconds per generation using RTX4090.

Table 1: Detection performance for local and global memorization at different inference steps $T$.

| | $T = 1$ | | | $T = 10$ | | | $T = 50$ | | |
| | AUC | F1 | T@1%F | AUC | F1 | T@1%F | AUC | F1 | T@1%F |
|---|---|---|---|---|---|---|---|---|---|
| Baseline (D) - Local | 0.918 | 0.864 | 0.629 | 0.989 | 0.982 | 0.953 | 0.990 | 0.983 | 0.560 |
| Ours (LD) - Local | **0.943** | **0.893** | **0.731** | **0.995** | **0.987** | **0.985** | **0.996** | **0.988** | **0.926** |
| Baseline (D) - Global | 0.979 | 0.944 | 0.934 | **1.000** | 0.987 | **1.000** | 0.999 | 0.976 | **1.000** |
| Ours (LD) - Global | **0.981** | **0.948** | **0.940** | **1.000** | 0.987 | **1.000** | 0.999 | 0.977 | **1.000** |

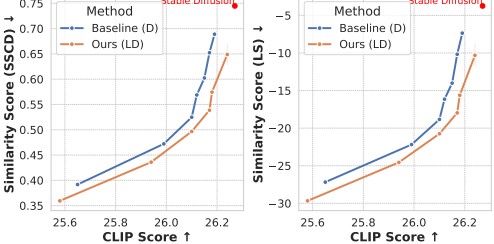

Figure 6: Local memorization's mitigation.

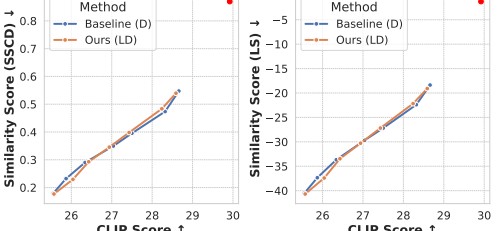

Figure 7: Global memorization's mitigation.

**Results.** As the baseline detection strategy in Eq. 7 incorporates a hyperparameter $T$, where the detection signal is defined as the average magnitude over $T$ inference steps, Tab. 1 evaluates performance across different values of $T$. This evaluation compares the baseline's magnitude with our proposed masked/localized magnitude, incorporating the BE mask as outlined in Eq. 8 under such different choices of baseline's hyperparameter values for demonstrating BE's ablation effects.

The results in Tab. 1 consistently demonstrate that integrating the BE mask improves detection performance under varying baseline hyperparameter settings across all metrics for local memorization while maintaining the performance for global memorization. These findings validate (1) the BE mask's effectiveness in accurately identifying memorized regions and (2) the importance of adopting a localized perspective for studying memorization, which remains under-explored. Additionally, Fig. 3 and 5 (Right) further demonstrate this targeted improvement in local memorization. The magnitudes for non-memorization and global memorization remain almost unchanged, but the magnitudes for local memorization are successfully amplified, making them more distinguishable from non-memorization cases. This effectively reduces the performance gap caused by local memorization by converting previous False Negative cases into True Positives (examples shown in Fig. 8).

For mitigation, we also include the original Stable Diffusion (SD) without any mitigation strategies applied in Fig. 6 and 7 for comparison, illustrating the extent to which we have reduced the local memorization performance gap. The horizontal axis (CLIP score) represents utility, while the vertical axis (similarity score) reflects privacy risks. More favorable privacy-utility trade-offs are indicated by points further toward the bottom right. This placement signifies lower privacy risks for the same utility level (same x-axis value) and higher utility for the same level of privacy risks (same y-axis value). Specifically, for local memorization, under the high-utility scenario where we aim to minimally reduce the CLIP score while mitigating memorization, our approach achieves a significantly better trade-off than the baseline. We reduce SD's similarity score by twice as much as the baseline while simultaneously observing a much smaller reduction in the CLIP score - less than half compared to the baseline. We also tested different privacy levels and consistently achieved a better trade-off for local memorization. Specifically, we observe a lower similarity score for the same CLIP score, as measured by both SSCD and our localized metric. Additionally, we consistently observe a higher CLIP score for the same similarity score. As expected, the results for global memorization remain almost identical to the baseline. This further confirms BE's effectiveness in addressing the performance gap introduced by local memorization in the task of mitigating memorization.

For evaluation metric, Tab. 2 demonstrates the superiority of our localized metric, LS (Eq. 9), which integrates SSCD, L2, and our BE mask, over the globalized metric, S (Eq. 10), which uses only SSCD and L2. This highlights the contribution of incorporating the BE mask. Furthermore, the results underscore the limitations of solely relying on SSCD, which performs well for global memorization but struggles with local cases. Notably, our design of LS, which incorporates SSCD, retains strong performance for global memorization while addressing the performance gap in local memorization, further validating the benefit of such a design.

It is worth noting that the effectiveness of these strategies not only demonstrates our improvement over the current state-of-the-art in existing tasks after incorporating BE's local memorization mask, but also serves as an extrinsic validation of BE's effectiveness in extracting local memorization masks. To our knowledge, BE is the first method to precisely extract local memorization masks in a highly efficient manner, requiring only one inference pass without the need to query the training data.

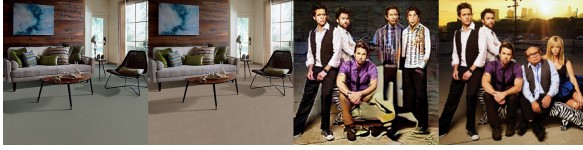

Figure 8: Examples of turning FN to TP via mitigation strategy. Generated image followed by training image.

Table 2: F1-score comparisons for different evaluation metrics.

|  | Local | Global |
|---|---|---|
| SSCD | 0.940 | 1.000 |
| S | 0.991 | 1.000 |
| LS (Ours) | **0.995** | 1.000 |

## 6 RELATED WORK

**Memorization in Diffusion Models** Numerous studies have previously explored memorization phenomena in language models (Carlini et al., 2021; Kandpal et al., 2022a;b; Lee et al., 2022) and GANs (Tinsley et al., 2021; Arora et al., 2018; Heusel et al., 2017). Over the past year, memorization within diffusion models has increasingly been scrutinized. Somepalli et al. (2023a) and Carlini et al. (2023) concurrently pioneered the examination of this issue, discovering that pretrained text-conditional models, such as Stable Diffusion, frequently replicate their training data. Similar findings were observed for unconditional generations in pretrained DDPMs on CIFAR-10 (Krizhevsky, 2009), as well as DDPMs trained on smaller datasets like CelebA (Liu et al., 2015) and Oxford Flowers (Nilsback & Zisserman, 2008). Additionally, Chen et al. (2024) explored class-conditional DDPMs on CIFAR-10, further highlighting the significance of this issue.

**Types of Memorization** Webster (2023) categorizes memorized captions into three types: (1) Matching Verbatim (MV), where the generated image using a memorized caption is an exact replica of the corresponding image; (2) Retrieval Verbatim (RV), where the generated image using a memorized caption replicates other training images rather than the corresponding one; and (3) Template Verbatim (TV), where the generated image using a memorized caption replicates only the template shared by a subset of training images, with variations in fixed locations outside the template, corresponding to different captions. This categorization focuses on the causes of memorization: MV results from overfitting the one-to-one mapping between image and caption in the training set, while many-to-many mappings lead to RV and TV. However, the concepts of Local Memorization (LM) and Global Memorization (GM) differ, focusing on whether the generated image exactly replicates an entire training image or just a portion of it, regardless of the causes of memorization. RV and TV can be seen as subsets of LM, as there are additional causes of LM beyond the "many-to-many mapping overfitting" that remain unidentified. For instance, in Fig. 8, the image in the third column is classified as MV by Webster (2023) due to the one-to-one mapping of the training image-caption, rather than many-to-many. However, it only memorizes a local portion of the image in the last column. This suggests that a pre-trained model may occasionally exhibit creativity when recalling overfitted memories, which can also lead to local memorization.

**Detection and Mitigation Strategies** Carlini et al. (2023) detects memorization by calculating the Euclidean distance between generated images and training images, recommending re-training on de-duplicated data as a mitigation strategy.Daras et al. (2023) suggests mitigating memorization by training models on corrupted data. Somepalli et al. (2023b) proposes to mitigate memorization by injecting randomness into text prompts, such as adding random tokens or noise, to break the link between memorized prompts and their corresponding images. Chen et al. (2024) employs a nearest neighbor search, similar to Carlini et al. (2023), to identify potential memorized outputs and then selectively applies guidance methods to instruct these generations away from the memorized training images, demonstrating outstanding mitigation performance. Recently, Wen et al. (2024) proposed an efficient method for detecting memorization during the inference phase of diffusion models. This approach uses the magnitude of text-conditional predictions for prompt detection, enabling high accuracy from the initial inference step, and triggers a mitigation strategy to optimize the prompt embedding when memorization is detected. Both the proposed detection and mitigation strategies have achieved state-of-the-art results. A most recent work (Ren et al., 2024) utilized cross-attention patterns to design detection and mitigation strategies, achieving performance comparable to Wen et al. (2024). The key insight is that the cross-attention scores for memorized prompts are less concentrated across all tokens in a prompt during multiple inference steps. This lack of concentration is quantified using entropy, which captures the distribution of cross-attention at the prompt-level.While both leverage cross-attention, it is important to note that our bright ending cross-attention (BE) differs from the cross-attention entropy approach regarding objectives, insights, inputs, and the final computed statistic. Specifically, different from Ren et al. (2024)'s objective of leveraging cross-attention for designing detection and mitigation strategies, BE also aims to achieve a new task of precisely and efficiently locating localized memorization regions. Also, BE does not rely on the concentration patterns of all tokens during multiple steps as the insight and inputs but instead uses the raw final step attention score of the end token. Also, BE employs the raw image-patch level distribution rather than the prompt-level distribution as the computed statistic, serving as a local memorization mask.

## 7 CONCLUSION

In this paper, we identify a performance gap in existing tasks of measuring, detecting, and mitigating memorization in diffusion models, particularly in cases where the generated images memorize only parts of the training images. To address this, we introduce a new localized view of memorization in diffusion models, distinguishing between global and local memorization, with the latter being a more meaningful and generalized concept. To facilitate investigation from a local perspective, we propose a new task: extracting localized memorized regions. We propose leveraging the newly observed 'bright ending' (BE) phenomenon to precisely and efficiently accomplish this task, resulting in a local memorization mask. Through extensive experiments, we demonstrate that the mask generated by this new task can also be used to enhance performance in existing tasks. Specifically, we propose a simple yet effective strategy to incorporate the BE local memorization mask into existing frameworks, refining them to adopt a local perspective. This approach reduces the performance gap associated with local memorization and achieves new state-of-the-art results. The improved results not only highlight our improvements in existing tasks but also serve as extrinsic validation of BE's effectiveness in the new task of local memorization mask extraction, underscoring the significant contribution of BE.

## 8 ACKNOWLEDGEMENT

This work was supported in part by the Australian Research Council under Projects DP210101859 and FT230100549.

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

# 9 APPENDIX

## 9.1 LIMITATIONS AND FUTURE WORK

One limitation of using the memorization mask extracted through the bright ending phenomenon in tasks like detection and mitigation is that it can only be observed at the end of the inference process. This requirement results in a longer processing time than existing strategies when applied to existing tasks such as detection and mitigation. However, we believe the additional time is justified by its significant positive contributions to investigating local memorization in diffusion models. Moreover, each inference takes only a few seconds on a consumer-grade GPU, making the process practical for real-world applications (elaborated in Sec. 9.2). While existing tasks are well-studied with established evaluation criteria, future work could explore how to evaluate the performance of the newly proposed local memorization region extraction task. Our current approach uses an extrinsic evaluation method that demonstrates improvements by incorporating the mask into existing strategies. Future research could also consider an intrinsic evaluation by directly comparing the extracted mask with ground truth masks. Another future direction could involve developing more advanced evaluation metrics for local memorization. In our work, we employed a straightforward approach by combining a masked version of the L2 distance with the widely adopted SSCD to compute similarity. Future research could focus on creating methods similar to SSCD, trained in a self-supervised manner but specifically tailored for more accurate localized copy detection.

## 9.2 EFFICIENCY OF THE BRIGHT ENDING (BE) APPROACH

The Bright Ending (BE) method demonstrates significant efficiency in its dual capabilities: localization of memorization regions and integration into existing memorization strategies.

**Localization Efficiency.** BE is the first method to accurately localize memorization regions in generated images without requiring access to training data or conducting computationally expensive nearest-neighbor searches over large datasets. Instead, it uses a single inference pass, which takes only a few seconds on consumer-grade GPUs. This streamlined approach ensures practicality for real-world applications and scalability for various scenarios.

**Integration into Existing Strategies.** In addition to its standalone capabilities, BE integrates seamlessly into existing memorization detection, mitigation, and evaluation frameworks, enabling state-of-the-art performance improvements. While this integration does require an additional inference pass (e.g., 50 denoising steps, approximately 2 seconds on an RTX-4090) to extract the BE mask, the trade-off is well justified by the method's significant contributions to addressing local memorization challenges.

**Insight into BE's Efficiency.** The BE phenomenon is rooted in the distinct behavior of the end token's cross-attention map during the denoising process. Non-memorized generations typically exhibit low-attention patterns at the end token in later steps, reflecting a shift from semantic focus to fine-detail refinement. In contrast, memorized generations follow an overfitted denoising trajectory, resulting in a distinct anomaly: the 'bright ending', which can be easily identified.

While the final-step BE is currently used for its robustness and clear anomaly distinction, leveraging BE at earlier inference steps is also feasible. Earlier-step BE could potentially reduce computational costs further, as the anomaly is observable throughout the denoising process, albeit less pronounced. Future work could explore the use of earlier-step BE to optimize the efficiency of the approach further.

**Justification for Final-Step BE.** The decision to focus on the final-step BE is supported by its demonstrated practical contributions and significant improvements in the detection, mitigation, and evaluation of local memorization. Despite the marginal increase in computational cost, the benefits in accuracy and utility outweigh the overhead. The final-step BE ensures that the method remains robust and effective while providing a reliable foundation for further advancements.

## 9.3 ILLUSTRATION OF THE LABELING PROCESS

As discussed in Sec. 5.2, current metrics often fail to accurately identify memorized cases, as shown in Fig. 2. Therefore, thresholding their reported similarity scores is not reliable for categorizing

generations into memorized and non-memorized cases to establish the ground truth for evaluating detection strategies. To address this limitation, we manually label the ground truth for each generated image. This section provides additional details about the manual labeling process to ensure reproducibility.

First, we download all ground truth images from Wen et al. (2024)'s official GitHub repository, where the data is systematically organized according to the memorization prompt dataset introduced by Webster (2023). Specifically, for each memorized prompt, the repository includes sub-folders containing training images prone to memorization. This may include a single image when the prompt consistently generates a unique replicate, or multiple images when the prompt leads to variations of replicated outputs. We then compare each generated image conditioned on a memorized prompt with the corresponding training images in the sub-folder. Each generation is labeled as one of three categories: global memorization, local memorization, or non-memorization. Examples of such training-generation image pairs used for labeling are shown in Fig. 9. The straightforward nature of this process enables quick and accurate labeling.

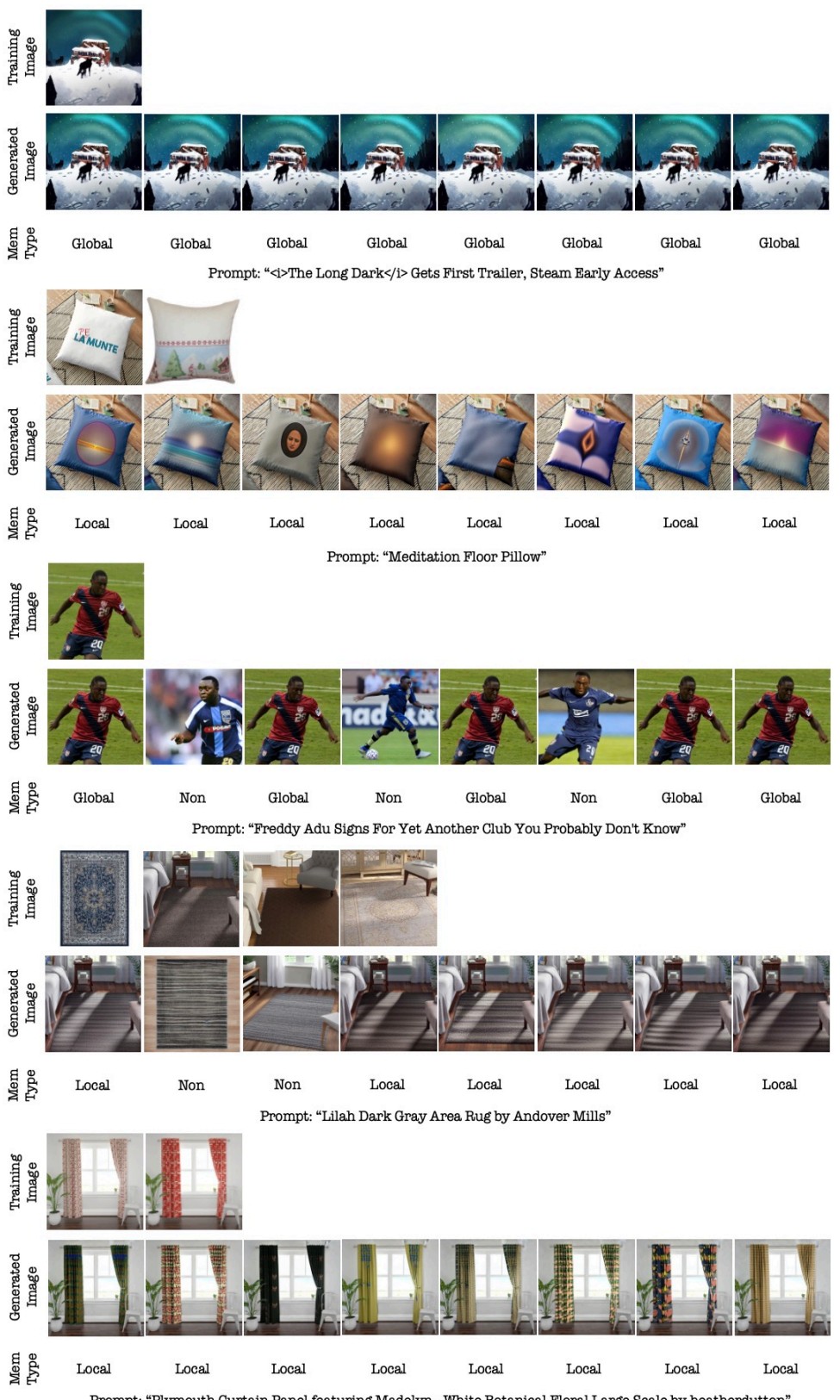

Figure 9: Examples of training-generation image pairs used for labeling memorization types.

