# OpenReview forum: "Exploring Local Memorization in Diffusion Models via Bright Ending Attention"
_ICLR.cc/2025/Conference — ICLR 2025 Spotlight_

### Official Review · Reviewer_8TzW · 2024-10-23

**Soundness:** 3
**Presentation:** 3
**Contribution:** 2
**Rating:** 6
**Confidence:** 2

**Summary:**

This paper introduces a novel perspective on memorization in text-to-image diffusion models by focusing on local memorization, where specific regions of a training image are memorized rather than the entire image. It identifies a unique "bright ending" (BE) anomaly in cross-attention patterns, where memorized regions show higher attention to the end token during the final denoising step. This insight enables the extraction of localized memorization regions without access to training data. The paper proposes a new task—locating these localized memorization regions—and integrates this approach with existing evaluation, detection, and mitigation strategies. The paper demonstrate that incorporating BE significantly improves performance in existing tasks, particularly in narrowing the gap caused by local memorization, achieving state-of-the-art results.

**Strengths:**

The paper introduces the concept of "bright ending" as a novel phenomenon in diffusion models, which has not been explored in prior research. This unique observation enables a new task of localized memorization detection, which broadens the scope of understanding memorization in generative models.

The paper is good written. The paper clearly differentiates between global and local memorization, offering detailed explanations of how BE is used for local region detection. Figures and visualizations of attention maps provide clarity on how the BE mechanism operates in practice, enhancing understanding of the proposed method.

The motivation of the paper is sound. It has implications for legal and ethical considerations, especially in cases where models might unintentionally replicate copyrighted content. The findings are particularly relevant for improving privacy-preserving techniques in model training and inference.

**Weaknesses:**

The efficiency of the proposed approach needs to be further demonstrated. The BE-based approach requires analysis at the end of the inference process, making it slower compared to some existing global methods that might detect memorization earlier in the process.

While the BE phenomenon is well-documented for text-to-image diffusion models, it is not clear how well this method would generalize to other types of diffusion models, like recently the flow-matching based models.

More evaluation metrics for local memorization could be explored for a comprehensive study.

**Questions:**

How does the proposed BE method perform in scenarios where memorization is more subtle or occurs in highly abstract image regions? Is there a threshold of memorization size or intensity where BE becomes less effective?

Is it possible for the method to be generalized to other types of generative models? Can the BE-based method be generalized to generative models that do not rely on text conditioning, such as unconditional image generation models? Can it be applied to video generation?

Given the increased inference time due to analysis at the final denoising step, do the authors have suggestions for optimizing the process to make it more efficient for real-time applications?

---

> ### Author Response · Authors · 2024-11-24
>
> **Response to W1 and Q3: Additional discussions and suggestions regarding efficiency.**
>
> We sincerely thank the reviewer for suggesting further demonstration of BE's efficiency and for the interesting question about optimizing the process for greater efficiency. These points are highly relevant and may interest many audiences.
>
> We wish to highlight BE's efficiency through its two main capabilities:
> - First, BE is the first method to accurately localize the memorization regions in generated images without requiring access to training data or performing a nearest-neighbor search over all ground truth images. Instead, BE requires only a single inference pass, taking just a few seconds on a consumer-grade GPU. This efficiency makes BE practical for real-world applications.
> - Second, BE can be seamlessly integrated into existing memorization detection, mitigation, and evaluation strategies, achieving new state-of-the-art results. As discussed in the "Limitations and Future Works" section, this integration may increase the computational cost compared to baseline strategies because BE requires an additional inference pass (50 denoising steps, approximately 2 seconds using an RTX-4090) to extract the mask before incorporating it into the strategies. However, we believe the additional computational cost is well justified by the significant contributions BE makes to studying local memorization in diffusion models.
>
> Regarding the reviewer’s question on optimizing efficiency, we are pleased to share that the 'bright ending (BE)' phenomenon is observable even in earlier inference steps, though it is not as pronounced as in the final denoising step. This stems from the same insight discussed in Section 4: as the denoising process progresses, non-memorized generations’ end tokens typically exhibit a dark cross-attention map containing only low-attention finer edges or scattered random points. Since the end token summarizes the semantic meanings of the prompt, this behavior indicates that, in later inference steps, the denoiser’s attention shifts from focusing on semantic meanings to concentrating on fine details as the image becomes cleaner.
> In contrast, memorized generations follow an inflexible overfitted trajectory, resulting in their end token’s attention scores displaying a distinct anomaly — the 'bright ending'. While leveraging BE from earlier steps could improve efficiency, the anomaly becomes most pronounced at the final step. Thus, we focus on the final-step BE to ensure robustness and effectiveness.
>
> Given the significant practical contributions demonstrated throughout this paper, we argue that the marginal increase in computational cost for using the final-step BE compared to earlier-step BE is well worth the benefits in accuracy and utility. We appreciate the reviewer for raising this insightful question, which we believe will interest a wide audience. To address this issue thoroughly, we have added Section 8.2 in the Appendix. Thank you for your valuable suggestion!
>
>
> **Response to W2 and Q2: Generalizability of BE to other types of diffusion models.**
>
> Thank you for this thought-provoking question, which has inspired us to reflect on BE’s broader implications. We believe the BE phenomenon, rooted in the distinct denoising patterns of overfitted generations, has potential generalizability to certain other diffusion models that have the following two attributes: (1) suffer from the memorization issue, (2) use a denoising framework.
>
> For video diffusion models, especially the text-conditional ones, BE could extend naturally by analyzing aggregated or sequential attention patterns across time, leveraging temporal consistency. For unconditional image generation models, it can be challenging to observe BE due to the lack of conditioning, but it may manifest as latent space anomalies or distinct noise prediction patterns that are worth investigating. However, for flow-matching-based models, which bypass denoising by learning trajectories between distributions, we are skeptical about BE’s applicability due to their fundamentally different design.
>
> We also observe that the degree of overfitting likely influences BE’s effectiveness. Text-conditional models, such as Stable Diffusion, are particularly prone to overfitting because text-conditioning acts as a "key" to the model's memory, creating more specific training data associations. Unconditional models lacking this specificity may exhibit less severe overfitting and, therefore, a subtler BE effect.
>
> This intriguing direction offers exciting avenues for future research, and we will investigate its feasibility in subsequent work. We appreciate the reviewer for bringing this valuable perspective to light!

---

> > ### Comment · Reviewer_8TzW · 2024-11-25
> >
> > Thanks the authors for the detailed response. I am actually more interested to see if the proposed method could be extended to other frameworks such as following matching, DiT, etc which are more mainstream now and to see the method could be generalized and has a wider audience. Therefore, I would like to maintain my initial score.

---

> > > ### Author Response · Authors · 2024-11-25
> > > **Thanks for the feedback!**
> > >
> > > We sincerely appreciate the reviewer’s thoughtful comments and interest in exploring the potential generalizability of our proposed BE phenomenon to other frameworks. This highlights exciting directions for future research, and we are grateful for the opportunity to further clarify the scope and significance of BE:
> > >
> > > - **Flow-matching-based models, while inspired by diffusion principles, are fundamentally distinct frameworks**. While flow-matching models represent a promising area, they remain conceptually and structurally different from traditional diffusion models such as they lack the explicit denoising process, and we believe it is natural for different frameworks to have customized designs that require non-trivial adaptions from methods designed for another framework.
> > >
> > > - **Our proposed method targets diffusion models that remain the mainstream framework and have a wide audience**. We wish to kindly clarify that this work targets the memorization issue in diffusion models, which directly appeals to the wide audience of all existing works in this domain and extends naturally to broader applications, such as video diffusion models. Furthermore, as diffusion models remain the mainstream framework for generative modeling, with active research exploring new conditioning mechanisms, noise schedules, and attention-based designs, we believe BE provides foundational insights that will inspire future innovations.
> > >
> > > - **The field of memorization in flow-matching-based models has not been pioneered yet**. To our knowledge, no existing work has reported memorization in flow-matching-based models. Therefore, validating the presence of memorization in these models would be a necessary first step before considering whether BE or its principles can be adapted.
> > >
> > > We greatly appreciate the reviewer’s insightful comments and agree that exploring the generalizability of BE to novel frameworks like flow-matching models is an exciting direction for future research. At the same time, we wish to emphasize that BE already addresses a wide audience within the mainstream diffusion model community and provides significant value in advancing the understanding, evaluating, detecting, and mitigating memorization in diffusion models.

---

> ### Author Response · Authors · 2024-11-24
>
> **Response to W3: Exploring more evaluation metrics.**
>
> We appreciate the reviewer for highlighting the potential to explore additional evaluation metrics beyond the first and only localized memorization metric introduced in this paper. While our proposed metric is simple and has room for further refinement, we agree that this represents a promising direction for future work. As noted in the "Limitations and Future Works" section, one potential strategy worth exploring is training self-supervised methods similar to SSCD. By tailoring the dataset preparation and training strategy specifically for localized copy detection, such approaches could enable model embeddings for masked images to compute meaningful cosine similarities with embeddings from other masked images. This would provide a more advanced way to integrate our localization insight with the robustness of SSCD's object-level similarity.
>
> We also greatly appreciate the reviewer's recognition of our multi-fold contributions, as noted in the "Summary" and "Strengths" sections of the review. These contributions include introducing a novel localized perspective on memorization by identifying performance gaps in existing strategies, pioneering the new task of locating localized memorization regions, uncovering the novel BE phenomenon for efficient and effective memorization mask extraction, and improving various tasks such as memorization detection, mitigation, and evaluation by simply integrating our BE mask. Among these, the improved evaluation metric represents just one of the benefits enabled by the BE mask. We are encouraged by the reviewer's acknowledgment and will build on these findings to propose more advanced evaluation metrics in future work. Thank you again for your recognition and valuable suggestions!
>
>
> **Response to Q1: Can BE's effectiveness be affected by thresholding choices of memorization size or intensity?**
>
> Thank you for this insightful question! BE's effectiveness is not impacted by the memorization size or intensity because its mechanism for incorporating the BE mask into existing strategies, such as memorization detection, does not rely on any thresholding hyperparameters. Instead, it solely depends on the pre-trained model's overfitted memory, making it robust and adaptable.
>
> Specifically, as discussed in Section 4, the bright ending (BE) attention map inherently serves as an automatically extracted memorization mask. This mask directly utilizes the BE attention scores as its values, eliminating the need for thresholding to convert it into a binary mask of only 0s and 1s. By doing so, it avoids the added variability introduced by thresholding hyperparameters, which may be less effective under certain threshold settings, thereby improving robustness. The approach remains effective regardless of whether the memorized regions are subtle or abstract, as it solely relies on the pre-trained model's overfitted memory to extract the mask. This eliminates the dependence on factors such as the degree of subtlety or abstraction of memorized regions or thresholding based on size or intensity of memorization. For a qualitative evaluation of the mask, please refer to Figures 1 and 4, where brighter patches indicate higher attention scores and darker patches indicate lower ones.
>
> We find addressing this question highlights a unique advantage of our method that was not previously emphasized in the paper. As a result, we have included additional explanations in lines 314-323 of Section 5.1 (highlighted in blue). We appreciate the reviewer's valuable contribution to improving the clarity of our work!

---

### Official Review · Reviewer_PxZo · 2024-11-03

**Soundness:** 3
**Presentation:** 3
**Contribution:** 4
**Rating:** 8
**Confidence:** 3

**Summary:**

This paper presents a problem with existing techniques to detect memorization in diffusion models in their lack to detect instances of memorization localized to specific subsections of generated images. To resolve this problem, they propose a new technique that creates a memorization mask for a generation.  The technique is based on a correlation between high attention values and memorization that the authors observe at the end of the diffusion process.

**Strengths:**

1. The problem presented by the authors is compelling, and they demonstrate failure cases of existing techniques in the literature
2. The authors propose a novel and effective method to address the problem
3. The proposed method seems to solve this problem while also maintaining the success cases of earlier techniques

**Weaknesses:**

1. Some of the figures and text are somewhat difficult to understand, especially relating to the method itself. For example, Table 1 and Figs 6/7 are not explained thoroughly enough, even though they seem to be central results for the paper. Section 5.1 could also benefit from further elaboration to ensure that the method and its formulation are clear.
2. The evaluation is not well explained in my opinion. For example, more details about the manual labelling process will make the results more reproducible.
3. Sharing the code and the dataset will help making the method more reproducible

**Questions:**

1. What is the motivation for incorporating SSCD similarity into the metric? Why not just use some function of the mask itself to generate a score?
2. Could you please share more details about the manual labeling process and the dataset itself?
3. Will you share the code implementing your method?

---

> ### Author Response · Authors · 2024-11-24
>
> **Response to W1: More explanation for Section 5.1, Table 1, and Figures 6/7.**
>
> Thank you for the reviewer’s valuable suggestion that Section 5.1, Table 1, and Figures 6/7 could benefit from further elaboration. We completely agree with this observation.
> In response, we have included detailed explanations in these three subsections. Specifically, the blue texts in the revised manuscript elaborate on Section 5.1 (lines 314-323), the explanation of Table 1 (lines 404-413), and the interpretation of Figures 6 and 7 (lines 421-425 and 431-433).
> We believe these updates significantly improve the overall presentation of the manuscript, making the paper easier to understand, and we sincerely appreciate the reviewer’s constructive feedback and contribution.
>
>
> **Response to W2 and Q2: More details about the manual labeling process and the dataset.**
>
> Thank you for pointing this out! We previously provided only a brief explanation of the motivation and feasibility for such manual labeling in lines 375–377 of Section 5.2, stating: "Due to current metrics' failure to accurately identify memorized cases, we manually label the ground truth for each generated image from the memorized prompts. Webster (2023) organizes the nearest training images for these generations, making it easy to observe local memorization cases with minimal subjectivity in manual labeling."
>
> The reviewer’s suggestion to include more details about the manual labeling process and the dataset is highly beneficial for ensuring reproducibility. To address this, we have added an additional sub-section (Section 8.1) in the Appendix, which provides a detailed illustration of the labeling process. This includes pointing out the sources from which we downloaded the datasets and qualitative examples in Figure 9 to demonstrate how the manual labeling was conducted. These additions highlight that the labeling process is straightforward and objective and allows for quick and accurate labeling.
>
>
> **Response to W3 and Q3: Sharing the code and dataset.**
>
> Thank you for this important question. We completely agree with the reviewer that open-sourcing these resources would significantly enhance the reproducibility of our work. To this end, we would like to assure the reviewers that we will release both the code implementing our method and the dataset used in our experiments upon the acceptance of this paper. This will include comprehensive documentation and example scripts to help the community easily replicate our results and apply our method to other tasks. We believe this commitment will make our contributions more accessible and beneficial to the research community.
>
>
> **Response to Q1: Motivation for incorporating SSCD similarity into the metric.**
>
> Thank you for this insightful question. The motivation behind incorporating both SSCD similarity and masked L2 distance into the metric lies in leveraging the strengths of both approaches. Masked L2 excels in its ability to localize memorization regions, providing pixel-level similarity/distance metrics, akin to traditional L2 distance. However, it remains constrained to pixel-level comparisons. On the other hand, SSCD quantifies object-level similarity, offering robustness against transformations such as scaling and shifting. Nevertheless, SSCD relies on embeddings from a pre-trained model designed for global image copy detection, making it less effective for local memorization scenarios, as illustrated in Figure 2 and discussed in Section 3. Additionally, SSCD cannot compute meaningful similarity scores for masked images.
>
> Our method integrates the localization capabilities of masked L2 with the robustness of SSCD's object-level similarity, effectively combining the benefits of both approaches. This simple yet effective integration allows us to achieve accurate evaluations across both global and local memorization cases, addressing the limitations of each individual metric.

---

> > ### Comment · Reviewer_PxZo · 2024-11-27
> >
> > Thank you for your detailed response!
> >
> > It addressed most of our concerns. Our main remaining concern is about the use of SSCD. We would like to validate that the difference between the methods presented in Eq.9 and Eq.10 is not just a result of suppressing the effect of the $\ell_2$ metric (which suffers from some biases), in favour of the SSCD metric. Could the authors please compare the results in Table 1 to just using the SSCD metric directly?
> >
> > Minor comments:
> > 1. It seems that Appendix 8.1 (or even 8) is not referenced from the main text.
> > 2. Could the labels in Table 1, and the figures, be made more meaningful than only “Baselines” and “Ours”; to indicate which is “LS” and which is “S”.
> >
> > Other than that, we think the revised manuscript is great!

---

> > > ### Author Response · Authors · 2024-11-27
> > > **Thanks for the feedback!**
> > >
> > > Thank you for your valuable feedback! We have submitted a revised paper version, which includes additional explanations and experimental results along with some minor refinements as kindly suggested by the reviewer, with all updates highlighted in purple for clarity.
> > >
> > > The main concern regarding Table 1 is highly insightful and highlights an area where our initial descriptions were insufficient. To clarify, the results in Table 1 do not rely on SSCD; they compare our detection strategy (LD in Equation 8) with the baseline detection strategy (D in Equation 7) rather than the evaluation metrics LS and S in Equations 9 and 10. This misalignment stems from an oversight in the Setup paragraph of Section 5.2, where we did not clearly map figures and tables to their corresponding tasks and equations.
> > >
> > > Thanks for the reviewer's valuable advice, in the revised manuscript, we have added detailed descriptions in lines 381-387 to explicitly link:
> > >
> > > 1. **Table 1** with Equations 7 and 8 (D and LD) and the task of memorization detection.
> > > 2. **Figures 6 and 7** with Equations 7 and 8 in the context of memorization mitigation.
> > > 3. **Table 2** with Equations 9 and 10 (LS and S) and the task of evaluating memorization, also including SSCD, to better demonstrate the effectiveness of our LS metric, thanks for the reviewer's inspiration. For a more detailed discussion of the comparisons, please navigate to our additions in lines 439-445 in Section 5.2 of the paper.
> > >
> > >
> > > For the two minor comments, we have correspondingly made the following updates:
> > >
> > > 1. Added references in lines 378–379 and 531 for Appendices 8.1 and 8.2, respectively.
> > > 2. Updated the notation in Equations 7 and 8 to "D" and "LD" for consistency with the experimental results.
> > > 3. Modified labels in Table 1 and Figures 6 and 7 to use clearer terms like "LD," "D," "LS," and "S" instead of the less descriptive "Baseline" and "Ours."
> > >
> > > We deeply appreciate the reviewer's valuable input, which has significantly improved the clarity and presentation of this work. Please feel free to share any further suggestions, as we would be delighted to address them!

---

> > > > ### Comment · Reviewer_PxZo · 2024-12-03
> > > >
> > > > We thank the authors for their thorough revision! We greatly appreciate the authors' additional text and have increased the score.
> > > >
> > > > We do have one further suggestion that could help make the paper an easier read for people who want to quickly understand the proposed method and results. Specifically, we feel that Table 1 being next to Figure 6/7 and using the same labels but lacking elaborate captions makes it unclear that one measures the efficacy of D/LD, and the other measures the efficacy of the mitigation strategy based on D/LD. We suggest that the authors either (1) split up Table 1 from Figures 6/7 such that they are closer to their respective paragraphs, or (2, preferably) move some of the content from the main text to the figure/table captions such that the figures and table may be thoroughly understood without help of the main text.

---

> > > > > ### Author Response · Authors · 2024-12-03
> > > > > **Thanks for your contributions!**
> > > > >
> > > > > Thank you once again for your insightful suggestions to further improve the presentation of our paper and for increasing the score! We greatly value both of your recommendations and will incorporate them into the revised version. Specifically, we will adjust the layout to present Table 1 and Figures 6/7 with separating paragraphs in between, rather than vertically aligned next to each other. Additionally, we will enhance the captions for both the table and figures by including additional details from the main text, ensuring they are more self-contained and easier to interpret without relying on surrounding text.
> > > > >
> > > > > We sincerely appreciate the time and effort you have dedicated to providing thoughtful feedback throughout the review process. Your contributions have been invaluable in enhancing the clarity and presentation of our work!

---

### Official Review · Reviewer_ve5E · 2024-11-03

**Soundness:** 2
**Presentation:** 2
**Contribution:** 2
**Rating:** 8
**Confidence:** 4

**Summary:**

This paper is a followup work of Wen et al. (2024), which discovered interesting patterns that "abnormally high predicted noise magnitude" indicate "global memorization". This work finds similar patterns between "abnormally high cross-attention values" of the EOS token in the prompt with the image tokens indicate "local memorization". The experiments validate such patterns can indeed be used to find local memorization.

**Strengths:**

1. The proposed idea is simple and is a natural extension of Wen et al. (2024).
2. The occurrence of "local memorization" has important practical values but is under-explored.

**Weaknesses:**

1. In multiple places, the authors state that the "bright ending" is obvious at the final denoising step. However, in the experimental results presented in Table 1, the 3 designs were using "First Step", "First 10 Steps", and "All Steps", respectively. Only "All steps" include "the final denoising step", but apparently "the final denoising step" plays only a minor role in "All steps". Why this discrepancy?
2. In section 5.1, "we propose element-wise multiplication of the magnitude by our memorization mask extracted via BE". However, the authors didn't give details about how to extract a memorization mask from BE in Section 4. Is this done using a fixed threshold? a dynamic threshold? Or a threshold predicted by a dedicated NN?

**Questions:**

1. AFAIK there's no dedicated EOS token used by stable diffusion. Instead, there are only 77-N padding tokens which are almost identical to each other (except for being added with different positional encodings). Suppose the prompt contains 15 tokens, then there will be 77-15=62 padding tokens which have identical token embeddings. I believe the cross-attention maps of these tokens are highly similar to each other. Do you compute the cross-attention map by averaging these 77-N attention maps? Or do you choose the last padding token to extract the cross-attention map?

EDIT after author response:
Apparently I got it wrong. There's indeed an EOS token inserted before padding tokens.

---

> ### Author Response · Authors · 2024-11-24
>
> **Response to W1: Clarification for Table 1.**
>
> Thanks for this valuable question that contributes to clarifying the purpose of Table 1.
> As the baseline detection strategy in Equation 7 incorporates a hyperparameter $T$, where the detection signal is defined as the average magnitude over $T$ inference steps, Table 1 evaluates performance across different values of $T$ (1, 10, and 50), which serves as the difference choices of hyperparameter values and is adapted from baseline's setup. This also allows for the demonstration of BE's ablation effects under different choices of the baseline's hyperparameter values.
> Specifically, under such design, Table 1's results can demonstrate the consistent advantage of Ours (magnitude with BE mask) over Baseline (magnitude without BE mask) under varying baseline hyperparameter settings across all metrics, validating (1) the BE mask's effectiveness in accurately identifying memorized regions and (2) the importance of adopting a localized perspective for studying memorization, which remains under-explored as you also kindly mentioned in the "Strength" section of your official review.
> Differently, the "step" in our description of the BE mechanism refers to its implementational detail of extracting such a BE mask. When testing its effectiveness in Table 1, it is already available and can be directly used in the computation in Equation 8 to obtain the masked magnitude.
>
> In summary, the "steps" in Table 1 and our description of BE serve different purposes; hence, using the exact wording can be confusing.
> We really appreciate the reviewer for pointing out this question, and we have correspondingly refined its presentation in our revised paper: (1) We have modified the headers in Table 1 to $T=1$, $T=10$, and $T=50$ to clarify that they only serve as a set of different hyperparameter choices adapting from the baseline's evaluation setting; (2) We have provided additional clarifications in lines 404-413.
>
>
> **Response to W2: Details regarding how to extract memorization mask from BE.**
>
> Thanks for this important question.
> The bright ending (BE) attention map itself inherently serves as the automatically extracted memorization mask.
> Specifically, it directly utilizes the BE attention scores as its values, eliminating the need for thresholding to convert it into a binary mask of only 0s and 1s. By doing so, it avoids the added variability introduced by thresholding hyperparameters. For a qualitative evaluation of the mask, please refer to Figures 1 and 4, where brighter patches indicate higher attention scores and darker patches indicate lower ones.
> Although we discussed in line 295 of Section 4 that the BE "effectively functions as an automatic memorization mask," and in lines 332–335 that "Unlike methods requiring hyperparameter tuning, such as setting thresholds to convert attention scores into a binary mask, our approach uses the attention scores directly as weights. This reweights the relative importance of each pixel in the magnitude computations," we sincerely appreciate the reviewer's valuable suggestion to include more details in Section 5.1, which can indeed improve this paper's flow. We have now included more explanations in Section 5.1. Please check out the blue text for the added details.
>
>
> **Response to Q1: Implementational details regarding EOS token in Stable Diffusion.**
> Thank you for the reviewer’s acknowledgment of Stable Diffusion's implementational details and for raising this question regarding its text tokenizer with concrete examples! This has made the question very specific and easy to address.
> We wish to clarify that the tokenizer provided by Stable Diffusion can, in fact, automatically handle the EOS token. Specifically, Stable Diffusion leverages the CLIP tokenizer, which automatically appends the <startoftext> (aka. SOS, token ID: 49406) and <endoftext> (aka. EOS, token ID: 49407) tokens during text tokenization. This ensures the model knows where the meaningful text starts and ends, and then either adds padding tokens or truncates to the required length of 77.
>
> For example, for a prompt containing $N = 15$ tokens, the decoded tokens will include <startoftext>, followed by the $N = 15$ tokens, then <endoftext>, and finally $77 - 1 - N - 1 = 60$ padding tokens. Our proposed bright ending (BE) strategy utilizes the cross-attention map from the <endoftext> token, as visualized in Figure 4.

---

> > ### Comment · Reviewer_ve5E · 2024-11-25
> > **Thanks for the response**
> >
> > This clarifies my questions. Thanks. Especially I realized that I had a misunderstanding about the EOS token in stable diffusion. Therefore I raised my rating to accept.

---

> > > ### Author Response · Authors · 2024-11-25
> > > **Thanks for the feedback!**
> > >
> > > Thank you for taking the time to carefully read our rebuttal and for raising your rating from 6 to 8. We sincerely appreciate your constructive feedback and insightful comments, which have greatly contributed to improving the clarity and flow of our paper.

---

### Meta-Review · Area_Chair_iE8q · 2024-12-19

**Metareview:**

The submission received the ratings of four reviewers, which recommended 8, 8 and 6, averaging 7.33. Given the plenty of competitive submissions in ICLR, this stands at a score above the acceptance. The reviewers' concerns mainly focus on some writing or presentation issues, which have been well addressed after the rebuttal. Therefore, I tend to recommend acceptance towards the current submission.

**Additional Comments On Reviewer Discussion:**

Well addressed about the writing and presentation issues.

---

### Decision · Program_Chairs · 2025-01-22

Accept (Spotlight)